# EMERGENT WORLD REPRESENTATIONS IN OPENVLA

## ABSTRACT

Vision Language Action models (VLAs) exhibit complex control behaviors without explicitly modeling environmental dynamics. However, it remains unclear whether VLAs implicitly learn world models, a hallmark of model-based RL. We propose an experimental methodology using embedding arithmetic on state representations to probe whether OpenVLA, the current state of the art in VLAs, contains latent knowledge of state transitions. Specifically, we measure the difference between embeddings of sequential environment states and test whether this transition vector is recoverable from intermediate model activations. Using linear and non-linear probes trained on activations across layers, we find statistically significant predictive ability on state transitions exceeding baselines (embeddings), indicating that OpenVLA encodes an internal world model (as opposed to the probes learning the state transitions). We investigate the predictive ability of an earlier checkpoint of OpenVLA and uncover hints that the world model emerges as training progresses. Finally, we outline a pipeline leveraging Sparse Autoencoders (SAEs) to analyze OpenVLA's world model. [1]

## 1 INTRODUCTION

Traditionally, RL methods fall into two categories: model-free, which learn policies directly from experience, and model-based, which explicitly learn environment dynamics to inform decision-making. Model-based RL typically involves learning a state transition function to explicitly predict future states (Sutton & Barto, 2018).

Previous works questioned the necessity of a world model representation to learn an optimal policy (Levine et al., 2020; Sutton, 1990; Schrittwieser et al., 2020). However, past literature has also theoretically entertained the notion that model-free RL might induce an emergent world model in the agent (environmental knowledge) (Wijmans et al., 2023; Francis & Wonham, 1976).

Vision-Language-Action models (VLAs) are transformers (Vaswani et al., 2017) trained to control, typically by behavioral cloning on expert data. They have shown promise in multiple real-world applications (Kawaharazuka et al., 2025). We focus on a state-of-the-art 7B-parameter model for robotics (Kim et al., 2024). Evidence of a latent world model would enhance trust in these systems, and be of interest to RL practitioners more broadly, as it would support the use of model-free RL.

The hallmark of model-based RL is a state transition function. We look for one by probing model internals (residual stream) using linear and non linear probes (Nanda et al., 2023; Li et al., 2023). In this context a state transition function can be expressed in additive terms using embedding arithmetic (Mikolov et al., 2013), which has seen adoption in the multi-modal context (Couairon et al., 2022), as embeddings of future states can be expressed in terms of addition between the embedding of the present and a state transition vector. Similar methods have shown emergent world models in simpler transformers applied to text-based games without explicit RL training (Li et al., 2023).

A limitation of probing is that probes can be correlational (Belinkov, 2022). This means that a probe's performance may tell us more about the probe than about the model. Hence, we compare the performance of probes trained on intermediate model activations with probes trained on the embeddings to establish a causal link between the model's computations and the presence of a world model. Model activations uniformly exceed the baselines, and prevail in statistical tests.

---

[1] The code to replicate **all** the experiments in the paper is available here: https://anonymous.4open.science/r/reproducibility-emergent-world-model-openvla-CCE1

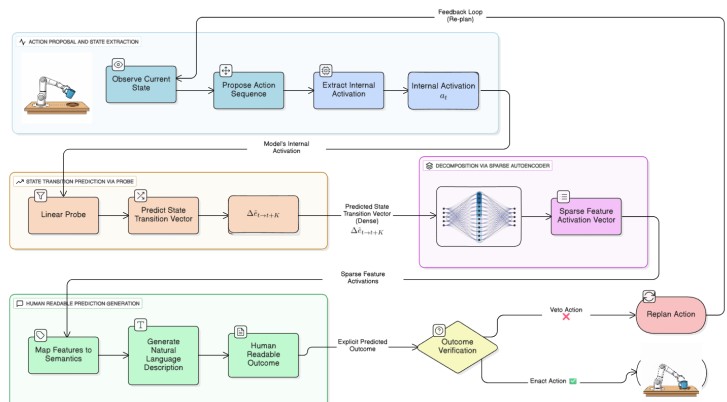

Figure 1: **An interpretable planning pipeline using the emergent world model of OpenVLA**. The pipeline intercepts an internal activation ($a_t$) from the model's policy. A linear probe predicts the resulting state transition vector ($\Delta\hat{e}_{t\rightarrow t+K}$), which is then decomposed by a Sparse Autoencoder (SAE) into a sparse vector of meaningful features. This enables human verification.

We investigate the effects of scaling training compute on the development of a world model, and we study its development across layers. Our contributions can be summarized as follows:

- We leverage embedding arithmetic to show that OpenVLA, which was trained by single-step behavioral cloning, has latent environmental knowledge by probing state transition vectors.

- We show that scaling pre-training compute enhances latent knowledge of state transitions, and we locate the world model across layers.

- We outline an application of Sparse Autoencoders to interpretable planning in which predicted state transitions extracted from the latent world model of OpenVLA become interpretable expectations to instruct veto or action execution, see Figure 1.

## 1.1 RESEARCH QUESTION: ARE ACTIVATIONS A BETTER WORLD MODEL THAN EMBEDDINGS?

OpenVLA represents each observation with a CLIP based embedding $\mathbf{e}_t$, which we treat as the model's environment state, and processes this through a transformer policy that produces internal activations $\mathbf{a}_t$ in the residual stream. Both $\mathbf{e}_t$ and $\mathbf{a}_t$ are potential carriers of world knowledge, but they differ: embeddings encode the current scene, while activations reflect intermediate computations that may integrate information about how the scene will evolve under the policy.

We therefore study prediction of the state transition vector

$$\Delta\mathbf{e}_{t\rightarrow t+K} = \mathbf{e}_{t+K} - \mathbf{e}_t$$

using probes trained either on embeddings $\mathbf{e}_t$ or on activations $\mathbf{a}_t$, across horizons $K \in \{1, 3, 10, 30\}$ and all *LIBERO* sections. This leads to the following research question:

> **RQ:** Do probes trained on internal activations $\mathbf{a}_t$ outperform probes trained only on embeddings $\mathbf{e}_t$ at predicting the state transition vector $\Delta\mathbf{e}_{t\rightarrow t+K}$?

If, under a fixed probe family, probes on $\mathbf{a}_t$ consistently achieve higher predictive power than probes on $\mathbf{e}_t$, the most parsimonious interpretation is that OpenVLA's residual stream does not merely store a static state, but computes a representation in which future state transitions are more directly accessible, consistent with an internal world model.

## 2 BACKGROUND

### 2.1 OPENVLA

We study OpenVLA (Kim et al., 2024), a 7B-parameter state-of-the-art vision-language-action model. It is trained on the Open X-Embodiment dataset (Collaboration et al., 2023) with behavioral cloning (Kumar et al., 2022), where the observations are single-step visual inputs and textual instructions on the task, and the actions are the end-effector position. We focus on all 4 subsections of the *LIBERO* dataset (Liu et al., 2023), which OpenVLA is evaluated and fine-tuned on. The sections are: *goal*, *spatial*, *long (10)*, and *object*, which encompass a wide variety of tasks. In particular, we focus on 400 episodes (100 episodes from each subsection) totaling 66931 steps across an early checkpoint (v01), OpenVLA, and fine-tunes to study the emergence of a world model.

### 2.2 EMBEDDING ARITHMETIC

Transformers across modalities (Couairon et al., 2022) leverage *vector arithmetic*, notably semantic directions and vector addition, to meaningfully compose and manipulate information. In Language Models, word embeddings reside in a geometry where *directions represent semantic axes* (e.g., gender, tense); adding vectors can analogously shift meanings (Mikolov et al., 2013).

### 2.3 MODEL-BASED AND MODEL-FREE RL

RL methods are traditionally divided into model-based and model-free approaches (Sutton & Barto, 2018). Model-based RL explicitly learns an environment model, typically a state transition function and a reward function, enabling the agent to simulate future trajectories and plan actions prior to execution (Moerland et al., 2022). This can yield high sample efficiency but requires accurate dynamics estimation, which can be challenging in complex or partially observable environments (Wang et al., 2019; Ha & Schmidhuber, 2018).

Model-free RL, in contrast, aims to find an optimal policy via interaction with the environment, without learning explicit dynamics. These methods often produce smooth, stochastic policies well-suited for continuous control but rely on trial-and-error, which is less sample efficient.

### 2.4 PROBING EMERGENT WORLD REPRESENTATIONS IN A SYNTHETIC BOARD GAME

Outside of RL, previous work has used probes to show that a transformer trained to predict legal moves in a simple board game (Othello) developed an emergent internal representation of the board state (Nanda et al., 2023; Li et al., 2023). This was done by probing the internal activations of the model to predict the state of the board after the *current* action of the model (its move).

### 2.5 KOOPMAN'S OPERATOR

In this subsection, we provide the basic concepts and definitions of Koopman's operator theory, which we use to study how to probe the world model of OpenVLA from its activations during an episode in 3.3 and 3.4. Koopman operator theory was first introduced in (Koopman, 1931), and in what follows we will partly adopt the notation of (Korda & Mezić, 2018). Consider a dynamical system

$$x_{t+1} = F(x_t)$$

where $F : \mathcal{X} \longrightarrow \mathcal{X}$ with $\mathcal{X}$ a topological space. Suppose we are given data points $\mathbf{X} = [x_1, \cdots x_M]$ and $\mathbf{Y} = [y_1, \cdots y_M]$ where $y_i = F(x_i)$ for $i = 1, \cdots M$. We call *observable* any map $g : \mathcal{X} \longrightarrow \mathbb{R}$ in $L^2(\mu)$ where $\mu$ is an invariant probability measure with respect to the environment dynamics, i.e. $\mu = \mu \circ F^{-1}$. Define the *Koopman operator* as the linear operator $\mathcal{K} : L^2(\mu) \longrightarrow L^2(\mu)$ s.t.

$$\mathcal{K}g = g \circ F.$$

Consider now a set of linearly independent observables $\{\psi_i \in L^2(\mu), i = 1, \cdots N\}$ and let $\mathcal{F}_N = \text{span}(\{\psi_i \in L^2(\mu), i = 1, \cdots N\})$. Define the *EDMD Koopman estimator* $\widehat{\mathcal{K}}_{N,M} : \mathcal{F}_N \longrightarrow \mathcal{F}_N$ as

$$\widehat{\mathcal{K}}_{N,M}\phi = c^\top A_{N,M}\Psi$$

for any $\phi \in \mathcal{F}_N$, where $A_{N,M}$ is the solution to the following least squares problem

$$\min_{A \in \mathbb{R}^{N \times N}} \|A\Psi(\mathbf{X}) - \Psi(\mathbf{Y})\|_F^2$$

with $\Psi(\mathbf{X}) = [\Psi(x_1), \cdots \Psi(x_M)]$, $\Psi(\mathbf{Y}) = [\Psi(y_1), \cdots \Psi(y_M)]$ and $\Psi(x) = [\psi_1(x), \cdots \psi_N(x)]^\top$. We then define the $L^2(\mu)$-projection of a function $\phi \in L^2(\mu)$ onto $\mathcal{F}_N \subset L^2(\mu)$ as

$$\Pi_N^\mu \phi = \arg\min_{f \in \mathcal{F}_N} \|f - \phi\|_{L^2(\mu)} = \arg\min_{f \in \mathcal{F}_N} \int_{\mathcal{X}} |f - \phi|^2 d\mu.$$

Finally, we define the *projected Koopman operator* as $\mathcal{K}_N = \Pi_N^\mu \mathcal{K}_{|\mathcal{F}_N}$ where $\mathcal{K}_{|\mathcal{F}_N}$ is the restriction of the Koopman operator to $\mathcal{F}_N$. Note that all the above can be restated for a generic time horizon $K$. We define the $K$-step Koopman operator $\mathcal{K}^K : L^2(\mu) \longrightarrow L^2(\mu)$ as

$$\mathcal{K}^K g = g \circ F^K,$$

where $F^K$ denotes the $K$-fold composition of $F$ with itself. Thus, $\mathcal{K}^K$ propagates observables forward by $K$ time steps, i.e.,

$$(\mathcal{K}^K g)(x_t) = g(x_{t+K}).$$

Notice that $\mathcal{K}^K = (\mathcal{K})^K$, which reflects the semigroup property of the Koopman operator.

## 3 METHODOLOGY

### 3.1 APPROXIMATING WORLD MODELS WITH KOOPMAN OPERATOR

We study the dynamics induced by a fixed (e.g., optimal) policy, and analyze the resulting *closed-loop* system in the embedding space:

$$e_{t+1} = F(e_t), \qquad e_t \in \mathcal{X}.$$

where $e_t \in \mathcal{X}$ is the embedding at time step $t$. We would like to show such a system (the state transitions induced by an optimal robot arm) *could* be extracted by probing OpenVLA's activations (as we do in 3.3 and 3.4).

We also assume that it exists an invariant measure $\mu$. In the following we fix a time horizon $K$ and, following the notation of the previous section, we consider $x_t = e_t$ and $y_t = e_{t+K}$. Moreover, as in (Korda & Mezić, 2018), we make the assumption that $(F, \mathcal{X}, \mu)$ is ergodic and that the samples $x_1, \cdots x_M$ are the iterates of the dynamical system starting from some initial condition $x \in \mathcal{X}$. Let $a_t = z(e_t) \in \mathbb{R}^N$ the activations at layer 15 of the policy model, and set $\psi_i = z_i$ and hence $\mathcal{F}_N = \text{span}(\{e \longrightarrow z_i(e)\}_{i=1}^N)$.

**Theorem 1.** *Suppose that the following assumptions hold*

(A) *The basis functions $\psi_1, \cdots, \psi_N$ are such that*

$$\mu(\{e \in \mathcal{X} \mid c^\top \Psi(e) = 0\}) = 0 \quad \text{for all nonzero } c \in \mathbb{R}^N \text{ and all } N$$

*($\mu$-independence).*

(B) *The Koopman operator $\mathcal{K}$ is bounded in the sense that*

$$\sup_{f \in L^2(\mu), \|f\|=1} \|\mathcal{K}f\| < \infty.$$

(C) *The observables $\psi_1, \cdots, \psi_N$ are part of an orthonormal basis of $L^2(\mu)$ for all $N$.*

*Fix $K \in \mathbb{N}$. Then for every $g \in L^2(\mu)$,*

$$\left\|(\widehat{\mathcal{K}}_{N,M}^K \Pi_N^\mu - \mathcal{K}^K)g\right\|_{L^2(\mu)} \leq \underbrace{\left\|(\widehat{\mathcal{K}}_{N,M}^K - \mathcal{K}_N^K)\Pi_N^\mu g\right\|_{L^2(\mu)}}_{\textit{estimation on } \mathcal{F}_N}$$

$$+ \underbrace{\left\|(\mathcal{K}_N^K - \mathcal{K}^K)\Pi_N^\mu g\right\|_{L^2(\mu)}}_{\textit{finite-basis error}}$$

$$+ \underbrace{\|\mathcal{K}\|^K \left\|(I - \Pi_N^\mu)g\right\|_{L^2(\mu)}}_{\textit{projection truncation}}.$$

*Moreover:*

- *(Estimation) For each fixed $N$, $\lim_{M \to \infty} \left\| \left( \widehat{\mathcal{K}}_{N,M}^K - \mathcal{K}_N^K \right) \Pi_N^\mu g \right\|_{L^2(\mu)} = 0$ for all $g \in L^2(\mu)$.*

- *(Basis) For any fixed $g \in L^2(\mu)$, $\lim_{N \to \infty} \left\| \left( \mathcal{K}_N^K - \mathcal{K}^K \right) \Pi_N^\mu g \right\|_{L^2(\mu)} = 0$ and $\lim_{N \to \infty} \left\| (I - \Pi_N^\mu) g \right\|_{L^2(\mu)} = 0$.*

*Consequently, for every fixed $g \in L^2(\mu)$,*

$$\lim_{N \to \infty} \lim_{M \to \infty} \left\| \left( \widehat{\mathcal{K}}_{N,M}^K \Pi_N^\mu - \mathcal{K}^K \right) g \right\|_{L^2(\mu)} = 0,$$

*i.e., $\widehat{\mathcal{K}}_{N,M}^K \Pi_N^\mu$ converges strongly to $\mathcal{K}^K$ on $L^2(\mu)$.*

Proof of Theorem 1 can be found in Appendix J. Under the assumptions of the previous theorem, letting $M \to \infty$ and $N \to \infty$, we recover $K$-step evolution of any fixed observable $g$ in $L^2(\mu)$ with arbitrarily small mean-square error. In other words, letting $m_K(e) = (\mathcal{K}^K g)(e)$ and considering any model-free $K$-step regressor $\hat{f}_M$ trained to predict $g(e_{t+K})$ from $e_t$, then it satisfies $\hat{f}_M \to m_K$ as $m \to \infty$ and, for fixed $g$, we have

$$\| \mathcal{K}_{N,M}^K \Pi_N^\mu g - \hat{f}_M \|_{L^2(\mu)} \longrightarrow 0$$

i.e., on the closed-loop stationary distribution $\mu$, model-based and model-free $K$-step predictions coincide in mean-square. Using the policy's activations as features is a pragmatic heuristic that tends to reduce the basis error by providing task-relevant coordinates; however, as we use a finite number of features in $L^2(\mu)$ there will be always a non zero representation error due to the finite basis approximation and $L^2$ projection. In practice we therefore expect nonzero error at finite $M, N$.

This means that for an optimal robotic arm, we might expect to model its state transitions by probing its activations, which motivates our later use of such probes in 3.3 and 3.4.

## 3.2 STATE TRANSITION VECTORS

Past literature has probed the environment itself out of model activations (Nanda et al., 2023; Li et al., 2023), but in the case of OpenVLA showing some latent knowledge of the environment is trivial (the scene is given, and does not change much as a result of a single action, for instance a slight change in robot arm orientation). On the other hand, perfect knowledge of the environment (as in of every pixel) would not be expected, as OpenVLA is not trained for that.

Therefore, we endeavor to demonstrate an emergent world representation by studying whether the model possesses *some* latent knowledge of a *state transition function*. In the context of OpenVLA (Kim et al., 2024), this takes the form of a function

$$f : \mathbf{a}_t \mapsto \Delta \mathbf{e}_{t \to t+K}$$

which, given the model's internal activations $\mathbf{a}_t$ at time $t$, predicts the change in the model's representation of the environment in embedding space $\Delta \mathbf{e}_{t \to t+K}$ resulting from the model's current actions after $K$ timesteps. We prove theoretical recoverability of future states from activations of an optimal agent in 3.1, but it is statistically more convenient to train the probes on labels where we subtract the present state from the future, as this directly links tests like $R^2 > 0$ and $p < 0.01$ to the hypothesis of predictive power over future states ($R^2 > 0$ on $e_{t+1}$ would be trivial, as it's very similar to $e_t$).

Since the model represents the environment in its embedding space, our objective resembles learning a state transition vector $\Delta \mathbf{e}_{t \to t+K}$, which, when added to the current embedding $\mathbf{e}_t$, yields the embedding of the environment at time $t + K$. Here, $K$ denotes the number of environment steps required for the model's actions to take effect, accounting for latency, the speed of the robotic arm, and the frequency at which observation frames are provided.

Following (Nanda et al., 2023) we investigate the case $K = 1$ (state immediately after an action), but since actions may not be instantaneous, as is the case, for example, in a synthetic board game (Li et al., 2023), we also look at cases $K = 3$, $K = 10$, and $K = 30$. Putting this formally, we seek:

$$\mathbf{e}_{t+K} = \mathbf{e}_t + f(\mathbf{a}_t)$$

where $f(\mathbf{a}_t) = \Delta\mathbf{e}_{t \to t+K}$.

Embeddings at each layer are averaged across token positions (image patches) using mean pooling, an established approach in vision transformers (Ko et al., 2022; Marin et al., 2023), resulting in a single vector representation. To study the location of OpenVLA's world model, we extract activations $\mathbf{a}_t$ at layers $7, 15, 22, 30$ from the residual stream for a given time step $t$, while target transitions $\Delta\mathbf{e}_{t \to t+K} = \mathbf{e}_{t+K} - \mathbf{e}_t$ are computed from embeddings $\mathbf{e}_t$.

### 3.3 LINEAR PROBES

Building upon our theoretical understanding 3.1. We train linear probes to predict the state transition vector $\Delta\mathbf{e}_{t \to t+K}$ from internal model activations $\mathbf{a}_t$ using Lasso regression, incorporating an $\ell_1$ penalty (Tibshirani, 1996). The probe is trained by minimizing the Lasso objective, which combines the mean squared error (MSE) with the $\ell_1$ regularization term:

$$\mathcal{L} = \|f(\mathbf{a}_t) - \Delta\mathbf{e}_{t \to t+K}\|^2 + \lambda \|\boldsymbol{\beta}\|_1 \tag{1}$$

where $f(\mathbf{a}_t)$ is the output of the linear probe, parameterized by weights $\boldsymbol{\beta}$, and $\lambda$ controlling sparsity.

To tune the hyperparameters (regularization strength $\lambda$ and learning rate), we perform a grid search, using a train/validation/test split to assess generalization (Kohavi, 1995). We train across all 4 subsections of the *LIBERO* dataset, and run all our evaluations on the respective test sets.

### 3.4 MLP PROBES

The Linear Representation Hypothesis (LHR) posits that LLMs linearly represent concepts in neuron activations (Park et al., 2024). Hence, we train both linear (Nanda et al., 2023) and Multi Layer Perceptron (MLP) probes (Li et al., 2023), to investigate whether the LHR applies in our setting.

### 3.5 EMBEDDING BASELINES

The main limitation of probes is that probing can be correlational (Belinkov, 2022). This means that a probe's performance may tell us more about the probe than about the model (in our case it may learn the state transition itself). To address this limitation, we train probes on raw embeddings as a baseline. This allows us to isolate the causal effect of OpenVLA's internal computations (through the skip connection) from the probe's predictive ability over the raw activations. In particular, we carry out one way statistical tests for the hypothesis: $R^2_{f(a_t)} > R^2_{f(e_t)}$ [2].

Where $R^2_{f(a_t)}$ (advanced probes) refers to the highest $R^2$ of probes (linear or MLP) being trained on activations or both activations and embeddings (for a given layer, $K$, and dataset). $R^2_{f(e_t)}$, our embedding (baseline) $R^2$ refers to the highest $R^2$ of probes (linear or MLP) trained on embeddings.

## 4 RESULTS

### 4.1 TEMPORAL COHERENCE IN EMBEDDINGS

We measure cosine similarity between scene embeddings $\mathbf{e}_t$ and $\mathbf{e}_{t+K}$ of OpenVLA to highlight temporal coherence across $K \in \{1, 3, 10, 30\} \forall 4$ *LIBERO* sections. $\forall K$, we compute similarity at each step $t$, average over episodes, and report the step-wise mean. In Appendix H we observe that: (1) similarity decreases smoothly as $K$ increases; (2) $\forall K, \forall$ datasets curves are similar in shape, and differ by a vertical offset proportional to $K$.

Embeddings that are closer in time (hence physical layout) are also closer in latent space, and vertical offsets hint at a consistent *time direction* over steps. This supports the temporal state transition vector

$$\Delta\mathbf{e}_{t \to t+K} = \mathbf{e}_{t+K} - \mathbf{e}_t$$

as meaningful and supports its use as a model-internal representation of temporal state transitions.

---

[2]The embeddings and the residual stream are quite different (see Appendix F), so passing this test is a higher bar than strictly necessary: $a_t$ may have lower average predictive ability, but more for important video patches.

Table 1: Successful ($p < 0.01$) permutation tests / probes with $R^2 > 0$ across 4 datasets.

| Probe Type | K=1 | K=3 | K=10 | K=30 | Overall |
|---|---|---|---|---|---|
| L7 Linear | 4/4 | 4/4 | 4/4 | 4/4 | 16/16 |
| L7 MLP | 4/4 | 4/4 | 4/4 | 4/4 | 16/16 |
| L15 Linear | 4/4 | 4/4 | 4/4 | 4/4 | 16/16 |
| L15 MLP | 4/4 | 4/4 | 4/4 | 4/4 | 16/16 |
| L22 Linear | 4/4 | 4/4 | 4/4 | 4/4 | 16/16 |
| L22 MLP | 4/4 | 3/3 | 4/4 | 4/4 | 15/15 |
| L30 Linear | 1/1 | 4/4 | 4/4 | 4/4 | 13/13 |
| L30 MLP | 3/3 | 4/4 | 4/4 | 4/4 | 15/15 |
| **Total** | 28/28 | 31/31 | 32/32 | 32/32 | **123/123** |

## 4.2 PROBING PERFORMANCE

Probe performance is evaluated using regression scores (Montgomery et al., 2012)[3]. We use $R^2$ as opposed to classification metrics (Nanda et al., 2023; Li et al., 2023) because both the action space and the environment (embeddings) are continuous (as opposed to discrete board states).

### 4.2.1 OUTPERFORMING BASELINES

The results (Figure 2) indicate that probes based on the model's internal activations $\mathbf{a}_t$ exhibit statistically significant predictive power for the state transition vector $\forall K$ (all $R^2$ confidence intervals are $> 0$, all p-values $< 0.01$ for at least one layer). Full results in Appendix B ($R^2 s \pm \sigma$). As expected, larger values of $K$ are associated with higher $R^2$, we discuss the reasons for this in 4.6. These findings empirically support emergent latent world models in VLAs.

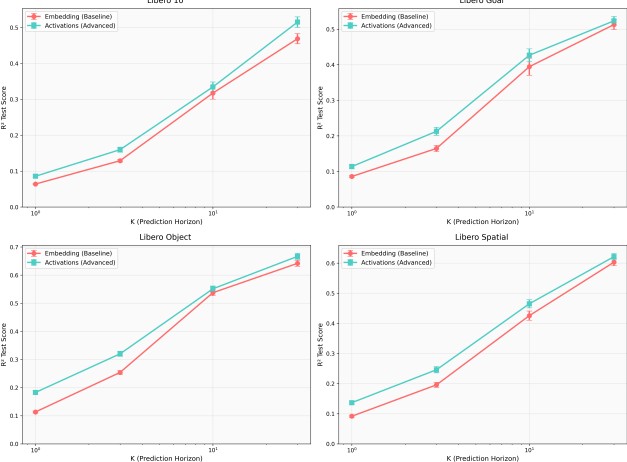

Figure 2: $R^2 \pm \sigma$ of the best probe trained on embeddings vs activations of OpenVLA across Ks and datasets. **Takeaway**: activations uniformly exceed embeddings, and most of the C.I.s don't overlap, underscoring statistical significance.

---

[3]We calculate confidence intervals across confidence levels with standard error across moving block bootstrap $R^2 s$ with $n_{\text{reps}} = 400$ replicates and automatic block length $b = \max(2, \lfloor n^{1/3} \rfloor)$ with Bessel's correction.

### 4.2.2 PERMUTATION TESTS

Statistical significance is assessed using permutation tests with 100 random label shuffles to calculate p-values (Good, 2000)[4]. We test only probes that achieve a positive $R^2$, and find that **all our permutation tests succeed**. Note that while we report a non-0 p-value following (Wasserstein & Lazar, 2016), none of the permutation tests we ran showed predictive ability. We ran tests for 123 probes with $R^2 > 0$ on OpenVLA, giving an **overall p-value** $< 0.0001$.

### 4.3 EMERGENCE OF WORLD MODELS ACROSS TRAINING

Scaling laws have been key to the success of transformers (Hernandez et al., 2021; Tay et al., 2022; Kaplan et al., 2020); hence, we investigate the effects of scaling training compute on the development of a world model by comparing predictive ability between OpenVLA and both an early checkpoint and fine-tuned models on all subsections of *LIBERO*.

We find that scaling (pre-)training compute (on Open X-Embodiment) is beneficial to the development of a world model (measured on *LIBERO*), as we find little evidence of a world model in v01 (early OpenVLA checkpoint). We look for a world model in fine tunes of OpenVLA, and we find that directly training on *LIBERO* yields less compelling evidence of a world model, which highlights the importance of pretraining for generalization. Full plots in Appendix A. These findings are consistent with the bitter lesson (Sutton, 2019), which argues that leveraging large amounts of data and compute enables models to develop advanced capabilities, whereas narrow task-specific data provide only limited gains.

### 4.4 LINEAR PROBES OUTPERFORM MLPS

We find some evidence in favor of the LHR by means of a 2-way statistical test, as MLP probes never have significantly higher $R^2$ scores than linear probes. This supports our use of SAEs in 5.2.

Table 2: MLP vs Linear Probe Performance Comparison

| Method | MLP Wins | Tie | Linear Wins |
|---|---|---|---|
| Absolute | 6/48 (12.5%) | 0/48 (0.0%) | 42/48 (87.5%) |
| 90% Two-Sided CI | 0/48 (0.0%) | 34/48 (70.8%) | 14/48 (29.2%) |
| 95% Two-Sided CI | 0/48 (0.0%) | 35/48 (72.9%) | 13/48 (27.1%) |
| 99% Two-Sided CI | 0/48 (0.0%) | 39/48 (81.2%) | 9/48 (18.8%) |

### 4.5 LOCATION OF WORLD MODELS

We investigate where OpenVLA may be computing its world model, and hence train probes across layers (7,15,22,30). We observe that knowledge of state transitions is best probed from middle layers. Later layers prove less effective at probing state transitions, in line with prior evidence that deeper layers encode logit-related information (Ghilardi et al., 2024). Full 3d plotting of $R^2$ across Ks and layers for OpenVLA on each dataset in Figure 3. See Appendix G for v01 and fine tunes.

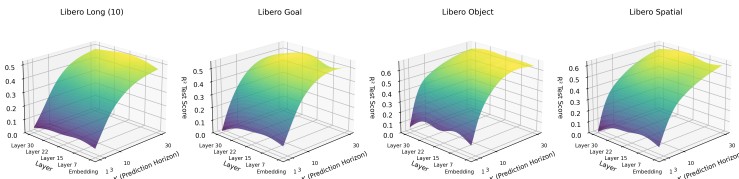

Figure 3: Test $R^2$ across layers and $Ks$ (with interpolation). Takeaway: **the world model is concentrated in the middle layers of OpenVLA.**

---

[4]Due to computational constraints, we run permutation tests with hyperparameters from the original dataset, leaving further tests and per-permutation tuning for future work.

### 4.6 ALLAN VARIANCE OF LONG TERM STATE TRANSITIONS

Allan variance (Allan, 1966) has been adopted in robotics to characterize noise and signal in sensors such as gyroscopes and accelerometers (Bhardwaj et al., 2015; El-Sheimy et al., 2008). We apply it to state transition vectors. Noise dominates at $K = 1$, and signal increases as $K$ increases (Appendix D), mirroring what happens in real-world sensors. The performance of our probes reflects this, with $R^2$ increasing as $K$ increases.

## 5 PRACTICAL ANALYSIS

### 5.1 SPARSE AUTOENCODERS

SAEs enhance the interpretability of LLMs by writing (superpositioned (Elhage et al., 2022)) neuron activations as a linear combination of interpretable sparse features (Bricken et al., 2023) reducing superposition (Huben et al., 2024). SAEs have recently been applied in the mechanistic interpretability of LLMs (Marks et al., 2024) and have been scaled to GPT4 (Gao et al., 2024). Zaigrajew et al. (2025) introduce the Matryoshka Sparse Autoencoder (MSAE) and apply it to CLIP embeddings (used by OpenVLA). The MSAE applies several Top-K operations with progressively larger values of $k$ to capture both coarse and fine-grained features, and produces multiple latent representations (for each $k$) reconstructing the input from each of them.

### 5.2 THE INTERPRETABILITY PIPELINE

Evidence for a linear world model in OpenVLA motivates a practical pipeline for interpretable planning. The central idea is to translate the model's intended action into an explicit and human readable prediction about how the environment representation will change, then decide whether to execute the action based on those predicted consequences.

1. Train a probe on a middle layer activation to predict the state transition vector $\Delta\mathbf{e}_{t\to t+K}$.
2. Obtain a decomposition of $\Delta\mathbf{e}_{t\to t+K}$ in terms interpretable representations from an SAE.
3. Locate the change in features across video patches. For instance, a feature corresponding to a mug might decrease in activation in the patch representing the table and increase in the patch representing the arm when OpenVLA selects actions to pick up the mug.
4. Check that the predicted plan is desirable and consistent with the proposed action. In high stakes settings (ex. surgical assistance (Wah, 2025)) this may inform approval or a veto.

SAEs applied to CLIP embeddings (which OpenVLA uses) yield interpretable features (Zaigrajew et al., 2025), and we have demonstrated the viability of probing state transitions. Hence, there remains for us to illustrate how SAE features in OpenVLA may be localized to specific patches, and while mean pooling prevents us from localizing features of $\Delta\mathbf{e}_{t\to t+K}$ directly (see 6), we prove (in Appendix I) that state transition patches live in the embedding space as embedding patches. This means that, given non-mean-pooled probes, the same method could locate features in $\Delta\mathbf{e}_{t\to t+K}$.

## 6 LIMITATIONS

A central conceptual limitation of our work is that all our evidence for an emergent world model is obtained via probing, which is inherently correlational (Belinkov, 2022). In principle, a sufficiently expressive probe could learn the environment dynamics from scratch and our results would then say more about the probe than about OpenVLA. We mitigate this concern in three ways, although we cannot eliminate it entirely.

First, **we compare probes with the same architecture trained either on embeddings $\mathbf{e}_t$ or on activations $\mathbf{a}_t$.** Under a fixed probe family, the consistent gap $R^2_{f(a_t)} > R^2_{f(e_t)}$ suggests that dynamics are easier to decode in the residual stream than in the raw embedding basis. Since residual stream activations combine the embedding with learned residual updates through skip connections, the additional predictive power must come from the computations performed by OpenVLA, not from additional probe capacity.

Second, our **Koopman analysis** in Section 3.1 **shows that**, under standard assumptions, **a linear map on a rich set of observables can approximate the** $K$ **step evolution operator in** $L^2(\mu)$. The fact that simple linear probes on $\mathbf{a}_t$ outperform linear and MLP probes on $\mathbf{e}_t$ is therefore naturally interpreted as OpenVLA having already transformed its representation into a coordinate system where the Koopman dynamics are closer to linear, i.e., where a world model is easier to read out.

Third, **the effect we observe is highly structured**: predictive power is concentrated in middle layers, improves with pre training compute, and is weaker in early checkpoints and narrow fine tunes. This pattern matches the picture that intermediate layers implement task relevant state evolution while later layers become more logit focused, and is harder to explain if the probes were the primary locus of the learned dynamics.

Nevertheless, our results should be understood as strong evidence that OpenVLA computes an implicit world model in its activations.

A limitation of our work is mean pooling: we can't unpack our predicted state transition vectors to do things like interventions or patch-level interpretability. This is a very exciting direction of future work. However, it would require substantially more compute than what was available for this study.

Another limitation of our work is that training was done on a total of 400 episodes which could put MLP probes at a disadvantage (as the increased parameter count requires more data). Moreover, epochs could be scaled further. However, we find that our train $R^2$s are higher than the test $R^2$s, which is evidence against under training for probes.

Moreover, we do not train an SAE, which is notoriously computationally intensive (Gao et al., 2024). Training an SAE would be the most important future direction for this work. However, retain that while we illustrate a potential application, fully investigating the application of SAEs to CLIP embeddings, similar to Zaigrajew et al. (2025), would constitute a distinct study.

## 7 CONCLUSIONS

We provide evidence that OpenVLA, through policy-based RL, develops an implicit world model for predicting future state transitions. Our findings show that internal network activations contain more predictive information than raw embeddings, with $p < 0.0001$ overall.

Key results include: (1) the world model emerges in the middle layers; (2) pre-training on diverse datasets is essential; (3) linear probes outperform MLPs, supporting interpretable representations; and (4) predictive performance improves with longer time horizons.

These insights suggest that model-free VLAs may be more robust than previously thought, capable of structured learning without explicit supervision. Our interpretability pipeline could enhance transparency in robotics, and future work should focus on training methods and exploring similar models in other domains. Overall, our findings blur the line between model-free and model-based reinforcement learning, showing that large-scale policy training can yield implicit world models as an emergent property of scale and data diversity.

## 8 ETHICS STATEMENT

This work presents no apparent ethical issues. The research does not involve experiments with human subjects, animal studies, proprietary or sensitive data, or foreseeable risks in deployment. However, it is important to stress that great caution must be taken when AI systems are deployed in the real world in a way that impacts human lives, as AI models can make mistakes, and even in cases where they can be interpreted, the explanations may be inaccurate (neither OpenVLA, nor our probes, nor SAEs are lossless).

## 9 REPRODUCIBILITY STATEMENT

We share how to fully reproduce results. Hence, we outline details of how we trained our probes in Appendix C. We provide hyperparameters in Appendix B (with the full results). Moreover, the code

to replicate **all** the experiments in the paper is available here: `https://anonymous.4open.science/r/reproducibility-emergent-world-model-openvla-CCE1`.

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

## A  EARLY AND FINE TUNED PERFORMANCE

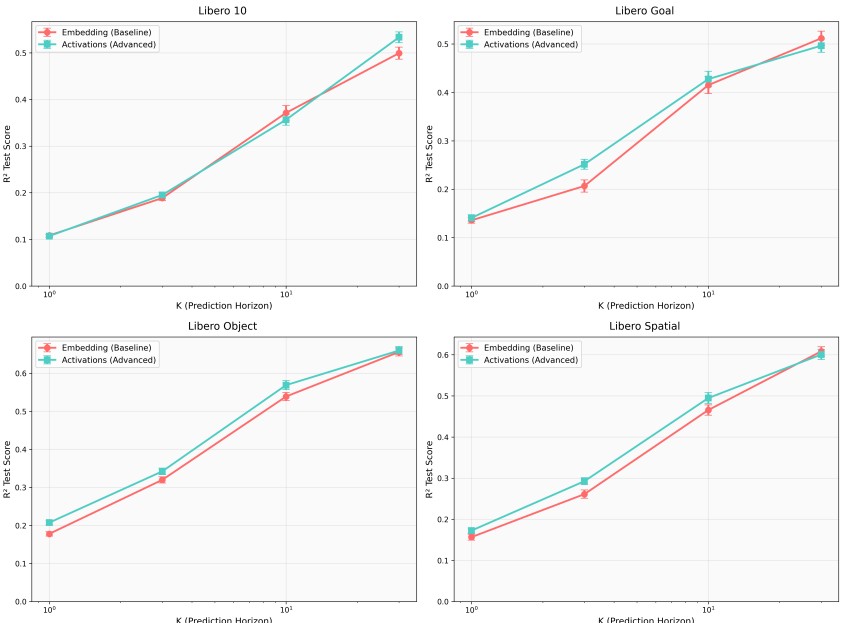

Figure 4: $R^2$ of probes trained on activations and embeddings of v01 (early OpenVLA checkpoint) across Ks and datasets. **Takeaway**: while activations often exceed embeddings, the embeddings also sometimes exceed activations. The model is overall less developed than OpenVLA's.

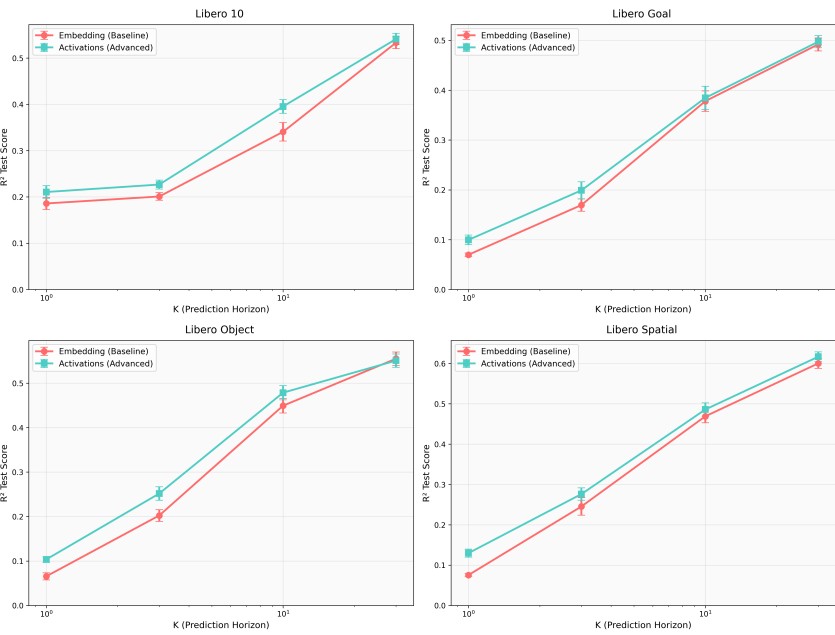

Figure 5: $R^2$ of probes trained on activations and embeddings of fine tunes of OpenVLA (on each subsection) across Ks and datasets. **Takeaway**: while activations often exceed embeddings, difference is not as wide as in the case of OpenVLA. The world model is slightly less developed than OpenVLA's.

Table 3: Comprehensive probe performance results for MAIN model

| Dataset | K | Train R² | Train Std | Test R² | Test Std | LR | Lambda | Dropout | Probe Type |
|---|---|---|---|---|---|---|---|---|---|
| long (10) | 1 | 0.1447 | 0.0014 | 0.0856 | 0.0039 | 1.00e-05 | 1.00e-09 | — | Linear-Regular-L7 |
| | 3 | 0.2558 | 0.0020 | 0.1597 | 0.0070 | 1.00e-05 | 1.00e-09 | — | Linear-Regular-L7 |
| | 10 | 0.4837 | 0.0038 | 0.3344 | 0.0137 | 1.00e-05 | 1.00e-08 | — | Linear-Joint-L15 |
| | 30 | 0.6759 | 0.0044 | 0.5151 | 0.0146 | 1.00e-05 | 1.00e-08 | — | Linear-Joint-L15 |
| goal | 1 | 0.1929 | 0.0019 | 0.1137 | 0.0066 | 1.00e-05 | 1.00e-09 | — | Linear-Regular-L7 |
| | 3 | 0.3486 | 0.0034 | 0.2128 | 0.0113 | 1.00e-05 | 1.00e-09 | — | Linear-Joint-L7 |
| | 10 | 0.5822 | 0.0054 | 0.4267 | 0.0182 | 1.00e-05 | 1.00e-08 | — | Linear-Joint-L15 |
| | 30 | 0.6781 | 0.0047 | 0.5234 | 0.0123 | 1.00e-05 | 1.00e-08 | — | Linear-Joint-L15 |
| object | 1 | 0.2381 | 0.0021 | 0.1827 | 0.0056 | 1.00e-05 | 1.00e-09 | — | Linear-Joint-L7 |
| | 3 | 0.4494 | 0.0030 | 0.3201 | 0.0089 | 1.00e-05 | 1.00e-09 | — | Linear-Joint-L22 |
| | 10 | 0.6613 | 0.0036 | 0.5512 | 0.0101 | 1.00e-05 | 1.00e-09 | — | Linear-Joint-L22 |
| | 30 | 0.7744 | 0.0032 | 0.6670 | 0.0110 | 1.00e-05 | 1.00e-09 | — | Linear-Joint-L22 |
| spatial | 1 | 0.2043 | 0.0020 | 0.1367 | 0.0054 | 1.00e-05 | 1.00e-09 | — | Linear-Regular-L7 |
| | 3 | 0.3563 | 0.0035 | 0.2460 | 0.0102 | 1.00e-05 | 1.00e-09 | — | Linear-Joint-L7 |
| | 10 | 0.5775 | 0.0052 | 0.4650 | 0.0133 | 1.00e-05 | 1.00e-08 | — | Linear-Joint-L15 |
| | 30 | 0.7145 | 0.0046 | 0.6214 | 0.0106 | 1.00e-05 | 1.00e-08 | — | Linear-Joint-L15 |

# B   FULL R2 AND P-VALUE TABLES

# C   REPRODUCIBILITY TABLE

We provide details for reproducibility in Table 5.

Table 4: Training hyperparameters and architecture configuration.

| | |
|---|---|
| Optimizer | Adam |
| Batch size | 512 |
| Epochs | 1000 |
| Final epochs | 300 |
| Grid sweep epochs | 50 |
| Seed | 0 |
| Data split | 70% train, 15% val, 15% test (chronological) |
| Residual stream dimensionality | 4096 |
| MLP layers | 2 |
| MLP hidden dim | 2x input size |
| Bias term in linear probes | not included |

Table 5: Training hyperparameters and architecture configuration.

# D   SIGNAL TO NOISE WITH ALLAN VARIANCE

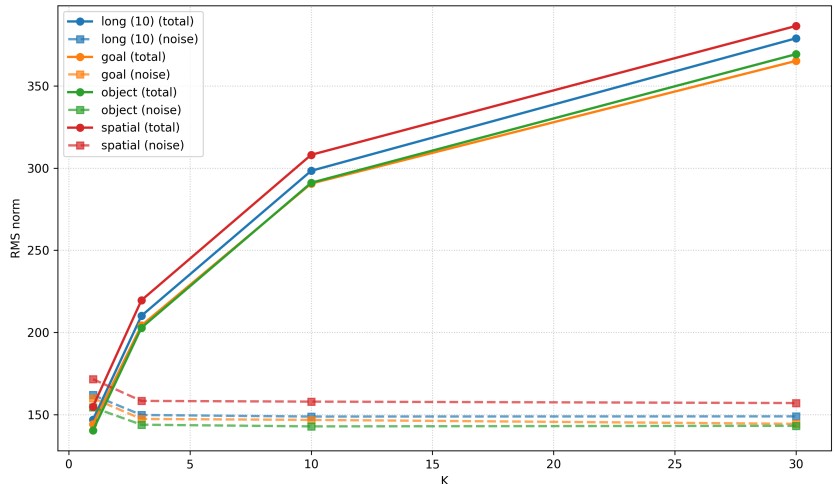

Figure 6: RMS (root mean square) total vs RMS noise contribution (all datasets).

# E   NEGATIVE RESULTS

We also report 2 negative results: scaling $K$ beyond 30 did not yield significant improvements (and as expected $R^2$ went down after around 100, as very long term changes are harder to predict). Moreover, using a window of of the past K steps instead of just one K steps ago did not yield any improvement (it led to overfitting), this is also expected as the model does not see past steps during training, and hence they are not used to build its world model (it remains an interesting theoretical lower bound, as our results constitute a lower bound for a sliding window of K steps, which would just be adding predictors).

# F   SIMILARITY BETWEEN ACTIVATIONS AND EMBEDDINGS

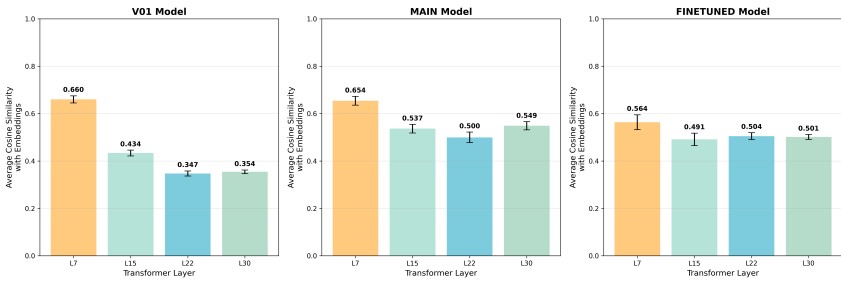

Figure 7: Cosine similarity between embeddings and activations across models layers and datasets.

## G  LOCATION OF WORLD MODELS FOR V01 AND FINE-TUNES

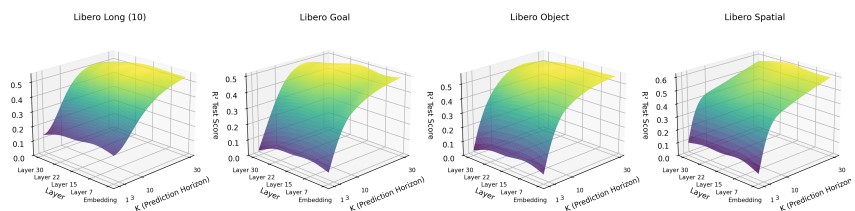

Figure 8: Test $R^2$ across layers and $Ks$ (with interpolation) for fine tunes.

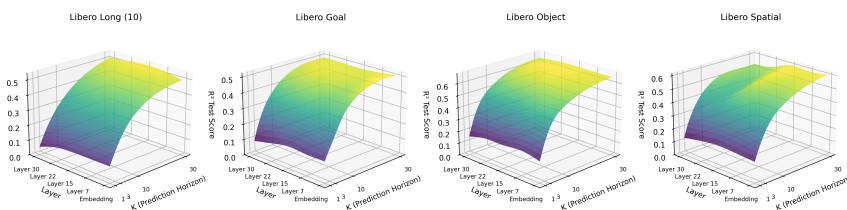

Figure 9: Test $R^2$ across layers and $Ks$ (with interpolation) for v01.

## H  TEMPORAL COHERENCE OF EMBEDDINGS

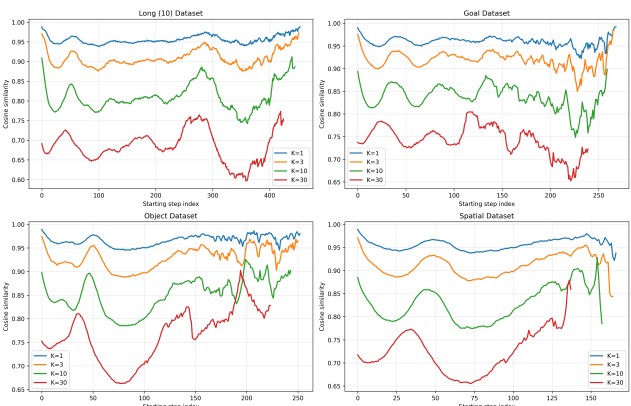

Figure 10: Cosine similarity between $\mathbf{e}_t$ and $\mathbf{e}_{t+K}$.

## I  PROOF THAT STATE TRANSITIONS LIVE IN EMBEDDING SPACE

**Claim.** Let $x_t$ be a video frame at time $t$. Let the CLIP vision encoder produce patch representations $\mathbf{p}_{t,1}, \ldots, \mathbf{p}_{t,N} \in \mathbb{R}^d$ and let the pooled image embedding be

$$\mathbf{e}_t := \frac{1}{N} \sum_{i=1}^{N} \mathbf{p}_{t,i} \in \mathbb{R}^d.$$

Then for any horizon $K \geq 1$,

$$\Delta \mathbf{e}_{t \to t+K} := \mathbf{e}_{t+K} - \mathbf{e}_t = \frac{1}{N} \sum_{i=1}^{N} \left( \mathbf{p}_{t+K,i} - \mathbf{p}_{t,i} \right).$$

In particular, $\Delta\mathbf{e}_{t\to t+K} \in \mathbb{R}^d$ lies in the same embedding space as $\mathbf{e}_t$, and it is a linear combination of patch level differences.

**Proof.** Mean pooling is a linear map $\mathsf{P} : \mathbb{R}^{d\times N} \to \mathbb{R}^d$ defined by

$$\mathsf{P}(\mathbf{P}) = \frac{1}{N}\sum_{i=1}^{N}\mathbf{p}_i, \quad \text{where} \quad \mathbf{P} = (\mathbf{p}_1,\dots,\mathbf{p}_N).$$

Let $\mathbf{P}_t = (\mathbf{p}_{t,1},\dots,\mathbf{p}_{t,N})$ and $\mathbf{P}_{t+K} = (\mathbf{p}_{t+K,1},\dots,\mathbf{p}_{t+K,N})$. By definition $\mathbf{e}_t = \mathsf{P}(\mathbf{P}_t)$ and $\mathbf{e}_{t+K} = \mathsf{P}(\mathbf{P}_{t+K})$. Using linearity,

$$\Delta\mathbf{e}_{t\to t+K} = \mathsf{P}(\mathbf{P}_{t+K}) - \mathsf{P}(\mathbf{P}_t) = \mathsf{P}(\mathbf{P}_{t+K} - \mathbf{P}_t) = \frac{1}{N}\sum_{i=1}^{N}(\mathbf{p}_{t+K,i} - \mathbf{p}_{t,i}).$$

Thus $\Delta\mathbf{e}_{t\to t+K} \in \mathbb{R}^d$ and is a linear combination of the patch differences, which establishes the claim.

$\square$

**Remark on linear projection heads.** If the pooled embedding is followed by a learned linear map $\mathbf{z}_t = W\mathbf{e}_t$ with $W \in \mathbb{R}^{m\times d}$ (no bias term), then

$$\Delta\mathbf{z}_{t\to t+K} = W\mathbf{e}_{t+K} - W\mathbf{e}_t = W(\mathbf{e}_{t+K} - \mathbf{e}_t) = \frac{1}{N}\sum_{i=1}^{N}W(\mathbf{p}_{t+K,i} - \mathbf{p}_{t,i}).$$

Hence differences after any weight-only linear projection are still linear combinations of patch-level differences. This directly supports our use of linear probes: the probe's predictions remain grounded in the same embedding geometry and can be traced back to interpretable patch-level changes when combined with an SAE.

## J  PROOF OF THEOREM 1

By triangular inequality, for every $g \in L^2(\mu)$, we have

$$\|\widehat{\mathcal{K}}_{N,M}^K\Pi_N^\mu - \mathcal{K}^K g\|_{L^2} \leq \|(\widehat{\mathcal{K}}_{N,M}^K - \mathcal{K}_N^K)\Pi_N^\mu g\|_{L^2} + \|(\mathcal{K}_N^K - \mathcal{K}^K)\Pi_N^\mu g\|_{L^2} + \|\mathcal{K}\|^K\|(I - \Pi_N^\mu)g\|_{L^2}.$$

By the ergodicity of the system and from assumption $A$, using Theorem 2 from (Korda & Mezić, 2018) we have consistency of the Koopman estimator:

$$\lim_{M\to\infty}\|\widehat{\mathcal{K}}_{N,M}^K\Pi_N^\mu - \mathcal{K}^K g\| = 0$$

where $\|\cdot\|$ can be any norm on $\mathcal{F}_N$.

Using assumptions $B$ and $C$, the conditions of Theorem 3 in (Korda & Mezić, 2018) are satisfied and hence we have

$$\lim_{N\to\infty}\|(\mathcal{K}_N^K - \mathcal{K}^K)\Pi_N^\mu g\|_{L^2(\mu)} = 0$$

For $g \in L^2(\mu)$, the $L^2(\mu)$–orthogonal projection onto $\mathcal{F}_N$ is

$$\Pi_N^\mu g = \sum_{j=1}^{N}\langle g,\psi_j\rangle\,\psi_j. \tag{2}$$

Parseval's identity gives

$$\|g\|_{L^2(\mu)}^2 = \sum_{j=1}^{\infty}|\langle g,\psi_j\rangle|^2. \tag{3}$$

Since $g - \Pi_N^\mu g \perp \mathcal{F}_N$, Pythagoras yields

$$\|g - \Pi_N^\mu g\|_{L^2(\mu)}^2 = \|g\|_{L^2(\mu)}^2 - \sum_{j=1}^{N}|\langle g,\psi_j\rangle|^2 = \sum_{j>N}|\langle g,\psi_j\rangle|^2 \xrightarrow[N\to\infty]{} 0. \tag{4}$$

Consequently, for any bounded operator $\mathcal{K}$ on $L^2(\mu)$ and any $K \in \mathbb{N}$,

$$\|\mathcal{K}\|^K\|(I - \Pi_N^\mu)g\|_{L^2(\mu)} \xrightarrow[N\to\infty]{} 0. \tag{5}$$

$\square$

# K    AUTOREGRESSIVE BASELINE

We further include a comparison between out activations' $R^2$ and an AR model as a baseline.

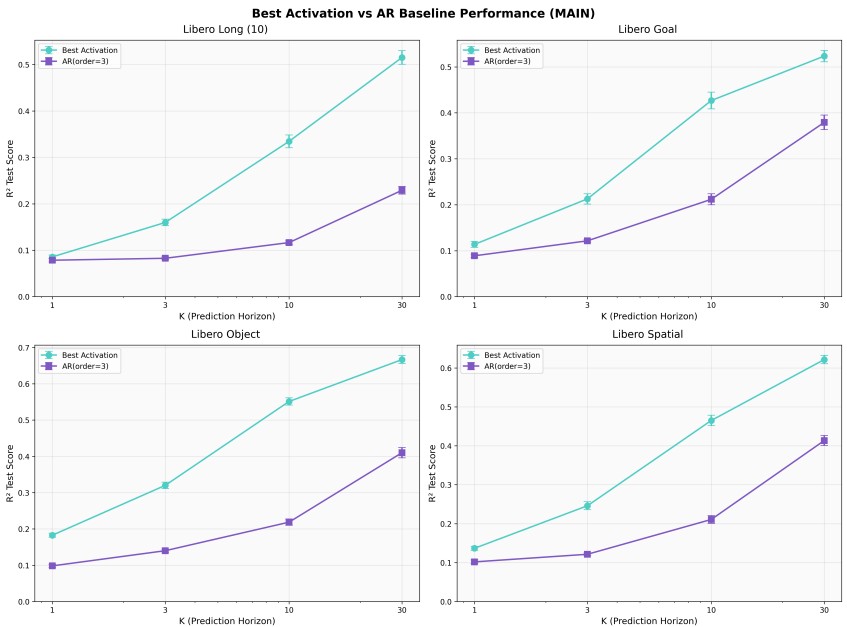

Figure 11: $R^2$ of activations vs an AR model. **we exceed the baseline by a large margin.**

# L    LLM USAGE STATEMENT

LLMs were used for help with writing and rephrasing some sentences/paragraphs in the paper, and to help with the latex formatting of some tables. They were also used to assist with coding.

