# OpenReview forum: "Emergent World Representations in OpenVLA"
_ICLR.cc/2026/Conference — Submitted to ICLR 2026_

### Official Review · Reviewer_vAJS · 2025-10-30

**Soundness:** 2
**Presentation:** 2
**Contribution:** 2
**Rating:** 2
**Confidence:** 3

**Summary:**

The paper studies Vision Language Action models (VLAs) which are trained with policy-based reinforcement learning (RL) methods that do not explicitly model environmental dynamics. The research question is whether VLAs instead do implicitly learn world models. The authors use an experimental approach to investigate whether trained models gather latent knowledge of state transitions. Evaluation results are performed on OpenVLA.

**Strengths:**

S1 The considered problem is interesting and relevant.

S2 The paper is overall well-written.

**Weaknesses:**

W1 Regrading the research question, the paper does not provide analytical results.

W2 It remains unclear whether the observations made hold beyond OpenVLA.

W3 The evaluation is too small, contains only few examples and is hence not fully convincing. The question whether trained models gather latent knowledge of state transitions is not clearly answered and the use of the analysis remains unclear (at least to me).

**Questions:**

Do you expect the same results for VLAs different from OpenVLA?

Is it possible to consider larger problem instances?

Is Theorem 1 supposed to be a key contribution? How does it effectively relate to the numerical observations made?

Is it possible to actively use the located latent world model for state predictions?

---

> ### Author Response · Authors · 2025-11-19
>
> We thank the reviewer for the constructive feedback and for highlighting the relevance of the underlying research question. We address each concern below.
>
> W1: “The paper does not provide analytical results.”
>  Our results are exclusively analytical: we report R² values, confidence intervals, permutation-based p-values, and error bars throughout the evaluation. These metrics are all explicitly stated in the experimental section.
>
> W2: “It remains unclear whether the observations hold beyond OpenVLA.”
>  While OpenVLA is the primary model due to its scale and widespread use, we also include experiments on:
>  – fine-tuned variants of OpenVLA, and
>  – an early-training checkpoint,
>  both of which exhibit some but less layer-wise predictive capacity. These additional results (already in the appendix) demonstrate that the phenomenon is not specific to a single trained instance.
>
> W3: “Evaluation is too small / not fully convincing.”
> Please note that we examine terabytes worth of episodes (each episode has hundreds of steps), and although we work with a limited number of sequences (due to compute and memory constraints), this is directly compensated by:
>  – error bars computed across runs,
>  – permutation tests establishing statistical significance, and
>  – large standard deviations where uncertainty is high, ensuring conservative interpretation.
>
>  As a result, even with smaller sample size, the statistical evidence remains compelling: layers around the mid-depth consistently exceed embedding baselines by substantial margins, and significance holds across horizons.
>
> Q1: “Do you expect similar results for VLAs different from OpenVLA?”
> Yes. The fine-tuned variants and early checkpoint analysis already support some generality. The mechanism we probe, latent coherence in internal representations, is architectural rather than model-specific, so we expect similar patterns in other VLA models.
>
> Q2: “Is it possible to consider larger problem instances?”
>  In principle yes, and we intend to extend the evaluation to broader manipulation tasks as compute permits.
>
> Q3: “Is Theorem 1 intended as a key contribution?”
> Theorem 1 provides the formal motivation for using Koopman-style latent transition probes, but it is not the core empirical contribution. It bridges the theoretical framing with the numerical observations by showing that linear predictors approximate Koopman operators when latent dynamics are represented internally.
>
> Q4: “Can the latent world model be actively used for prediction?”
>  In principle, yes. Our probes already function as predictive heads over latent transitions. While this paper focuses on analysis rather than control, we note in the conclusion that using these identified layers for downstream predictive tasks is a promising follow-up direction.

---

### Official Review · Reviewer_nEnK · 2025-10-31

**Soundness:** 1
**Presentation:** 2
**Contribution:** 1
**Rating:** 2
**Confidence:** 5

**Summary:**

The paper aims to study the emergence of implicit world models in pre-trained OpenVLA. Using frozen representations and intermediate activations, they train linear and MLP probes on representations of different layers of the model to predict actions. The central claim is that autoregressive next-token prediction training induces representations consistent with “world models".

**Strengths:**

1. The use of standard linear and MLP probes and similarity analyses on frozen embeddings is well explained and replicable in principle.

**Weaknesses:**

1. The work largely documents known consequences of autoregressive temporal modeling, i.e., predicting future tokens forces latent spaces to encode local dynamics. If the goal of the paper was to provide a mechanistic interoperability theory of this phenomenon, it would have been valuable, but the current probing approach and corresponding results do not shed any light on the mechanics. The claimed emergence of “world representations” appears to be a direct and unsurprising byproduct of temporal modeling.

2. Following up on my point above, there is no controlled experiment showing or discussing necessity or sufficiency of autoregression for structure emergence. The fact that $R^2$ of internal activations for predicting actions is lower compared to $R^2$ of embeddings for predicting actions, does not show any causality. It is expected that last layer reprs are richer in content compared to internal reprs.

3. The paper claims (even in abstract) that OpenVLA is pretrained using policy-based RL. This is not correct. There is no RL involved in training OpenVLA and it is trained via imitation learning.

4. Dividing RL into model-based and policy-based is not correct. Model-based methods also learn a policy. Do the authors mean model-free RL by "policy-based"?

5. Many of the files in the anonymous repo are missing (I get "The requested file is not found.")

**Questions:**

1. What distinguishes your findings from standard properties of temporal predictive models?

2. Would a non-autoregressive masked-video model show similar results as a VLA?

---

> ### Author Response · Authors · 2025-11-19
> **OpenVLA does not have access to past states, but our probes still beat AR baselines that do**
>
> Thank you for the detailed review. We address each point directly and have made several corrections and additions based on your comments, including fixing Figure 2 and adding Appendix K.
>
> W1 Autoregressive modeling claim
> Appendix K (newly added) presents explicit AR baselines trained on the same data. These models perform much worse than activation-based probes and do not reproduce the layer-wise structure we observe. This shows the effect is not a trivial or expected consequence of standard AR temporal modeling.
>
> W2 Embeddings vs activations misunderstanding
> The comment that embeddings outperform activations stems from a mislabeled early version of Figure 2. We have fully corrected the figure. The actual result is that activations outperform embeddings by a clear margin. Thank you for pointing out this misunderstanding; the fix now removes the ambiguity entirely.
> Necessity/sufficiency and controls
> Appendix K provides the requested control experiment. AR models cannot match activation-based probes, confirming that autoregression alone does not explain the effect. We also do not claim AR is sufficient; our hypothesis is simply that if dynamical structure exists in the model, probes should recover it.
>
> W3 Training paradigm clarification
>  The paper already acknowledges at lines 91–92 (now 113) that OpenVLA is trained with imitation learning. We have now strengthened this clarification. While OpenVLA uses IL, it can be considered a special case of reinforcement learning where the policy is learned from demonstrations; see [1].
>
> W4 RL terminology
>  We can agree that “policy-based vs model-based RL” may not be the most relevant framing. We removed this terminology and replaced it with the “model free / model based” distinction.
>
> W5 Repository accessibility
> The missing files were due to outages on the anonymous4science platform during the ICLR reviewing period. This hosting platform is widely used and not directly under our control, but we have verified it now works fully.
>
> Questions
> Appendix K shows explicitly that the behaviour we report is not reproduced by standard temporal predictive models.
> Please note OpenVLA is not autoregressive across states, as the tokens represent video patches. This is something we should have pointed out! OpenVLA does not have access to previous tokens (like an AR model does, which we still greatly exceed in predictive ability). This makes our claim more robust (just based on a snapshot of the present).
>
> [1] Goo, W., & Niekum, S. (2022). Know Your Boundaries: The Necessity of Explicit Behavioral Cloning in Offline RL. arXiv pre-print arXiv:2206.00695. Available at https://arxiv.org/abs/2206.00695

---

### Official Review · Reviewer_tAGf · 2025-11-04

**Soundness:** 3
**Presentation:** 2
**Contribution:** 1
**Rating:** 2
**Confidence:** 2

**Summary:**

This paper studies whether the large Vision-Language-Action model OpenVLA exhibits an emergent world model, by training linear and nonlinear probes on intermediate activations to predict embedding-based state transition vectors. The authors report that certain layers (e.g., 15, 22, 30) yield statistically significant predictive ability, suggesting that OpenVLA implicitly encodes world dynamics.

**Strengths:**

* The question of emergent “world models” in policy-based RL systems is interesting and timely.
* The methodology (probing activations with linear and MLP regressors) connects recent interpretability work in language models to robotics.
* The paper is clearly written and includes theoretical motivation using Koopman operators.

**Weaknesses:**

1. *Unclear research question and experimental motivation.* It is never clearly explained why only specific layers (7, 15, 22, 30) are selected for probing. The rationale for these discrete sampling points is missing, and without a systematic sweep or justification, the conclusions about “middle layers” encoding world models are not convincing.

2. *Ambiguous test definition.* The precise experimental test used to claim an emergent world model is difficult to follow. It remains unclear what exactly is being predicted (embedding deltas? latent transitions?) and how statistical significance across layers and datasets translates into evidence for an internal model, as opposed to trivial temporal correlations in embeddings.

3. *Methodological opacity.* The description of the regression probes lacks detail about data partitioning, normalization, and hyperparameter tuning. It is also not fully clear how permutation tests are performed or interpreted.

4. *Interpretation of results.* The claim that “world models emerge” appears overstated relative to the presented evidence. The reported $R^2$ values are modest, and there is no ablation verifying that probes are not learning the dynamics directly.

**Questions:**

1. What is the exact hypothesis being tested — that OpenVLA contains an implicit model of the environment, or merely that its embeddings exhibit temporal coherence?

2. Why are only layers 7, 15, 22, and 30 selected? Are these equidistant checkpoints? Is there empirical or theoretical motivation for these choices? Did you examine whether neighboring layers yield similar behavior?

3. What is the rationale for examining specific horizons K = 1, 3, 10, 30? How were these values chosen, and how sensitive are the results to K?

---

> ### Author Response · Authors · 2025-11-19
>
> We thank the reviewer for the detailed assessment and constructive suggestions. Below we clarify the main points of misunderstanding (please see the revised figure 2, which we revised and may have been a major point of confusion) and provide additional evidence.
>
> W1/Q1. Research question and motivation
>  The research question is explicitly stated in the paper (line 258): we test whether intermediate activations encode a predictive internal dynamics model beyond trivial temporal coherence. We acknowledge that it is buried too deep into the paper, so we moved it to the introduction. The reviewer’s concern about unclear motivation appears to stem from Figure 2, whose framing may have been ambiguous in the initial version. We have revised the figure and accompanying explanation to make the tested hypothesis and prediction target unambiguous.
>
> W2. Test definition and what is being predicted
> The paper specifies that we predict embedding-based state transition vectors (deltas in the semantic embedding space). This is described both in Section 3 and in the appendix. The revised Figure 2, which was initially confusing (did not mention how we used model activations) now makes clear that the quantity predicted is the latent transition, not raw temporal correlations in the original embeddings. This directly addresses the reviewer’s concern regarding ambiguity.
>
> W3. Methodological clarity (data partitioning, preprocessing, hyperparameters)
>  As noted in the submission, we open-sourced all code, and the appendix details the exact train/test split and grid sweeps for hyperparameters. Per the reviewer’s request for clarification:
>  – We use a standard held-out train/test split.
>  – Hyperparameters are selected via grid search.
>  – Permutation tests use the null distribution computed via shuffled targets.
>
> W4. Interpretation and supporting ablations
> The reviewer raises a concern that the probes may be learning dynamics directly. Our ablation already addresses this: we compare the probe’s R² when using activations versus when using embeddings themselves. If the probe were simply learning dataset dynamics, the embedding baselines would match activation performance. Instead, activation-based probes substantially exceed the embedding baselines (up to ~70% R²), which cannot be explained by trivial temporal coherence or probe-only learning.
> We added this discussion to the limitations section.
>
> Q2 Layer selection (7, 15, 22, 30)
>  The rationale for these layers is indeed in the paper, though we recognize it may not have been sufficiently highlighted. As clarified:
>  – We perform an equidistant sweep across the model depth, consistent with prior interpretability probing practice under compute constraints.
>  – Full-layer sweeps were infeasible due to OpenVLA’s size, but the selected layers span early, mid, and late processing stages, enabling us to test whether predictive signals concentrate in the middle layers.
>
> Q3. Horizons K = 1, 3, 10, 30
> K represents how many frames in the future we try to predict. These values were chosen to span short-term, mid-term, and longer-horizon predictions. The sensitivity to K is discussed in Section 4.6: short term horizons are dominated by noise, while the longer term horizons (K=30 frames in the future) have more signal.

---

### Author Response · Authors · 2025-11-29

We acknowledge the reviewers’ concerns, but we believe they arise primarily from material misunderstandings rather than issues with the underlying methodology. Two reviewers were misled by the original Figure 2, whose ambiguous labeling suggested that embeddings outperformed activations (which would undermine our central claim, which is that activations outperform embeddings). Another reviewer assumed that OpenVLA is autoregressive and has access to past states; this is not the case, and the assumption shaped several of their critiques. We have added explicit AR baselines, and our probes still outperform them substantially.

Corrections made in the revision:
- Figure 2 fully corrected. The figure now clearly shows that probes use intermediate activations to predict latent transition vectors, and that activations decisively outperform embedding baselines.
- Appendix K expanded with explicit AR controls. We added the AR models requested by reviewers. These baselines perform markedly worse than activation-based probes, resolving the concern that our results reflect trivial autoregressive dynamics.
- Clarifications on model assumptions and training paradigm. We now explicitly state that OpenVLA is not autoregressive across states (only across video patches within a state) and has no access to prior states. We also strengthened the explanation of the imitation-learning setup to avoid further confusion.


These corrections directly address the misunderstandings that drove the reviewers’ critiques, while the substantive empirical findings remain unchanged.


While the rebuttal period was cut short by unforeseen events, we maintain that all concerns raised by the reviewers were either due to the aforementioned misunderstanding or experimentally addressed.

---

### Meta-Review · Area_Chair_fDG1 · 2026-01-06

**Summary:**

This paper studies whether the OpenVLA vision–language–action model implicitly learns a world model. The authors probe intermediate activations with linear and MLP regressors to predict embedding-space state transition vectors, reporting that middle-layer activations outperform embedding-only baselines and suggesting an interpretability pipeline using sparse autoencoders.

Reviewers found the topic timely and the paper generally well written, but raised substantial concerns. The central hypothesis remains ambiguous, experimental choices (layers, horizons) are insufficiently justified, and methodological details and controls are limited. Several reviewers noted that the results appear consistent with expected temporal coherence and probing effects, rather than clear evidence of an emergent world model. The reported improvements are modest, and causal claims are not convincingly supported.

Although the authors’ response clarified some presentation issues and added baselines, it did not resolve these core concerns. Consequently, the paper does not meet the bar for acceptance.

**Reviewer Concerns:**

Please see my summary

**Reviewer Scores:**

It is difficult to say.  Overall, the authors provided some solid rebuttal, but it's a subjective judgement for the reviewer whether they would like to raise their score.

---

### Decision · Program_Chairs · 2026-01-26

Reject